# The Protective Effects of Sour Orange (*Citrus aurantium* L.) Polymethoxyflavones on Mice Irradiation-Induced Intestinal Injury

**DOI:** 10.3390/molecules27061934

**Published:** 2022-03-16

**Authors:** Zixiao Jiang, Zhenqing Li, Fengchao Wang, Zhiqin Zhou

**Affiliations:** 1College of Horticulture and Landscape Architecture, Southwest University, Chongqing 400716, China; k0054587541@126.com (Z.J.); zhenqinglee@126.com (Z.L.); 2State Key Laboratory of Trauma, Institute of Combined Injury of PLA, Burns and Combined Injury, Army Medical University, Shapingba, Chongqing 400038, China; 3The Southwest Institute of Fruits Nutrition, Banan District, Chongqing 400054, China

**Keywords:** *Citrus aurantium* L., bioactive compounds, polymethoxyflavones, acute radiation sickness, antiradiation agent, cytotoxicity

## Abstract

Sour orange (*Citrus aurantium* L.) is one of the biological sources of polymethoxyflavones (PMFs), which are often used to deal with gastrointestinal diseases. The intestine is highly sensitive to irradiation damage. However, limited certain cures have been released for irradiation-induced gastrointestinal injury, and the potentials of sour orange PMFs as radio-resistance agents have not been fully discussed yet. The present study aims to (1) investigate the PMF components in 12 sour orange cultivars, (2) determine the protective effects of PMFs on irradiation-induced intestinal injury by treating mice that received 12 Gy abdominal irradiation with different doses of PMFs and observing the changes in organ indexes and pathological sections and (3) test cytotoxicity of PMFs by CCK-8 method. The results showed that sour orange PMFs appeared to have high intraspecies similarity. Besides, PMFs protected mice from irradiation-induced injury by alleviating body weight loss, reliving organ index changing and maintaining the intestinal structure. Finally, IC_50_ concentrations to cell line CCD 841 CoN of PMFs and nobiletin were calculated as 42.23 μg/mL and 51.58 μg/mL, respectively. Our study uncovered PMF contents in 12 sour orange materials and determined the protective effects on irradiation-induced intestinal injuries, providing guidance for the utilization of sour orange resources.

## 1. Introduction

The intestinal tract is one of the most important immune organs and undertakes functions of digestion. The intestinal epithelium draws a clear distinction between internal and external environments of organisms and is composed of different cell types, including Paneth cells, intestinal stem cells (ISCs), endocrine cells, goblet cells, tuft cells, M cells and intestinal epithelial cells (IECs) [1,2]. The homeostasis of different cells contributes to nutrient uptake and pathogen resistance. Affected by complex biotic or abiotic factors, the intestinal epithelium developed a rapid self-renewal strategy to lessen the possibility of mutation and to maintain the function of the intestine. As most epithelial cells are well differentiated and lack proliferative capacities, the homeostasis of the intestine is mainly supported by ISCs located at the base of crypts [3,4]. Therefore, maintaining the stability of the stem cell pool is of great importance for intestinal functions.

The gastrointestinal (GI) form of acute radiation sickness (ARS) is a common disease often observed among patients who received intraperitoneal radiotherapy. Symptoms may appear in any section of the intestine and may be featured as abdominal pain, diarrhea (or obstruction), ulcer and apostasies [5,6]. The severity of the disease may be influenced by body weight, sexual distinction and chronic diseases. Loss of intestinal barrier function has been widely accepted as one of the major causes of ARS (GI form). Irradiation destroys the intestinal mechanical barrier function by promoting apoptosis of IECs and ISCs as well as blocking the expression of tight junction proteins (TJPs) [2,7,8,9]. Loss of ISCs will gradually change the cell composition, followed by a decrease in secretory cells which leads to a lack of chemical barrier function. Immune cells and microorganisms may be also affected by irradiation which causes the collapse of both the immune barrier and microbial barrier [10]. Approaches have been discovered to relieve the symptoms of ARS, including nutritional support, probiotic-preparation intervention, surgery, antiphlogistic drugs and hyperbaric oxygen therapy [5,6,9]. However, most of the cures mentioned above show limited benefits and have high risks of recurrence. Thus, effective and safe intervention is required for postoperative recovery of ARS.

Sour orange (*Citrus aurantium* L.), regarded as the crude materials of ‘zhiqiao’ and ‘zhishi’, has often been used in herbal medicine to deal with gastrointestinal diseases. A variety of bioactive compounds have been detected in sour oranges, such as flavonoids, volatile components (i.e., limonene, pinene, linalool, etc.) and alkaloids [11]. Among numerous compounds in sour orange, a group of flavones bearing two or more methoxy groups (also known as polymethoxyflavones (PMFs)) have been proven to have bioactivities of antioxidant, anti-inflammation and antifatigue [12]. Since the first PMF was identified in the 1930s [13], over 100 PMF monomers have been detected and identified. Extraction methods of common PMFs have been developed, making PMFs a group with relatively immense potential for functional food discovery. It has been confirmed that PMFs could help reduce inflammatory injuries in colitis as well as optimize intestinal barrier function by upregulating autophagy and expression of TJPs [7]. Recent studies also observed that PMF administration had impacts on gut microbiota in obesity and metabolic syndrome models [14]. Flavonoids, for example, β-naphthoflavone [15], were validated to protect cells from radiation-induced injury, yet the protective effects of PMFs on radiation-induced injury in vivo have not been fully discussed.

In the present study, PMFs in different sour orange cultivars were extracted and detected. PMFs in one germplasm, out of twelve cultivars, was extracted and purified for in vivo experiments. Different doses of purified PMF extracts were provided to mice in a consecutive 14-day treatment (7 days before irradiation and 7 days after irradiation). The PMF extracts were given through irrigation once per day, and mice were sacrificed on day 14 to evaluate the protective effects of sour orange PMFs on mice ARS. The results of our study uncover the composition of PMFs in sour orange cultivars and determine the potential of PMFs being a radio-resistant agent, providing practical guidance for the utilization of sour orange resources.

## 2. Results

### 2.1. Extraction and Detection of Polymethoxyflavones in Sour Oranges

Fruit materials of 12 sour orange cultivars were collected from Citrus Research Institute, SWU/CAAS. To determine the content of flavonoid compounds in collected sour orange materials, a library of 18 common flavonoids (Figure 1A) in citrus fruits was built for qualitative and quantitative analysis. The exact contents of monomers in each cultivar were calculated and summated to analyze the PMF distribution patterns in various sour orange cultivars from the component level. Eight polymethoxyflavones were detected in sour orange peels, and the content of the total polymethoxyflavones showed significant variation among different cultivars (Figure 1B). Cultivar ‘Lianhe’ contained the largest amount of PMFs in total, followed by cultivar ‘Yaogan’ and 10 other cultivars. In all PMFs detected, nobiletin was the most widely distributed monomer, while 4′,5,6,7-tetramethoxyflavone only existed in specific cultivars, such as ‘Yaogan’. To determine the ratio of PMF monomers in sour orange cultivars, a 100% stacked bar chart was generated to show the PMF distribution patterns from the cultivar level. Despite the differences observed in the total content of PMFs, the PMF monomers’ relative content distribution showed similarity that nobiletin (varied from 131.26 μg/g·FW to 7.84 μg/g·FW among cultivars), sinensetin (varied from 28.87 μg/g·FW to 1.01 μg/g·FW among cultivars), isosinensetin (varied from 63.92 μg/g·FW to 0.34 μg/g·FW among cultivars) and 4′,5,7-trimethoxyflavone (varied from 46.92 μg/g·FW to 2.91 μg/g·FW among cultivars) are four most widely distributed monomers in sour orange (Figure 1C). The similarity of sour orange PMFs ingredients was also verified by our hierarchical clustering and PCA results when introducing the PMF content of mandarin (*Citrus reticulata* Blanco cv. ‘dahongpao’, abbreviated as ‘DHP’), sweet orange (*Citrus sinensis* (L.) Osbeck. var. brasliliensis, abbreviated as ‘NHE’) and pomelo (*Citrus maxima* (Burm) Merr. cv. ‘liangpingyou’, abbreviated as ‘LPY’) as outgroups (detected by our previous studies [16]). Citrus materials were divided into three major groups by hierarchical clustering based on their contents of PMFs. All sour orange cultivars and pomelo cultivar ‘liangpingyou’ clustered together, while sweet orange and mandarin showed relatively larger distances (Figure 1D and Appendix A).

### 2.2. Purification of Sour Orange PMFs

Considering that the component of sour orange PMFs showed high similarity compared with other *Citrus* materials, we chose the type species (cultivar ‘daidai’), also known as a common crude material of Chinese traditional medicine ‘zhiqiao’ and ‘zhishi’, out of 12 sour orange cultivars for the following extraction and purification. The main flavonoids of the mixture after enrichment include two flavones, eight flavanones and eight polymethoxyflavones (Figure 2), including nobiletin (374.62 mg/g), tangeretin (251.74 mg/g), sinensetin (32.82 mg/g), 4′,5,6,7-tetramethoxyflavone (7.14 mg/g), isosinensetin (6.58 mg/g) and hexamethoxyflavone (1.53 mg/g), and small amounts of flavanones (or their glycosides), such as poncirin (14.79 mg/g). This result showed that the main component of PMFs in crude extract did not change after enrichment by macroporous resin, but the total content of PMFs increased from about 7.5% to over 60% (Figure 2A). In summary, these results demonstrated that the proportion of flavanone and flavone components (1.5~4.5 min) decreased and the relative proportion of PMFs (5.0~6.5 min) increased significantly after enrichment, achieving the purpose of separation and purification of PMFs (Figure 2B).

### 2.3. Potential Protective Effects of Sour Orange PMFs on Mice ARS

To determine the potential targets of ARS, we used the gene expression matrix of dataset GSE126507 in which samples were intestinal tissues collected from abdominal-irradiated (12 Gy) mice to mine the differentially expressed genes and pathways as the candidates of the potential target pool. Considering that estrogen might contribute to the radio-resistance, only male mice samples were extracted for the following analysis. Ionizing irradiation significantly changed the gene expression pattern in abdominal-irradiated mice compared to mice that received no irradiation (Figure 3A). The differential expression genes were annotated and GO/KEGG enrichment was carried out to analyze the potential pathways which were responsible for differences in intestinal damage repair after irradiation (Appendix A and Figure 3B).

The potential targets of PMFs were downloaded from online databases SWISS and SEA. The probability of PMF monomer acting as an antiradiation component was estimated preliminarily by calculating the counts of enriched pathways which met with those obtained from the public dataset. The potential pathways of PMFs matched well to pathways enriched by public sequencing data. PMFs showed the potential to have an impact on mice ARS as the targets downloaded were significantly enriched in all top three KEGG pathways (circadian rhythm, KEGG ID: mmu04710; NOD-like receptor signaling pathway, KEGG ID: mmu04621; and transcriptional misregulation in cancer, KEGG ID: mmu04621) in the candidate pool (Figure 3C). Protein–protein interaction (PPI) network of PMF targets and differential expression genes after irradiation were established by STRING which further revealed the possibility of PMFs as radioprotection agents (Figure 3D).

### 2.4. The Protective Effects of PMFs in Radiation-Induced Intestinal Injury Mice Model

To verify the radio-protective effects of PMFs, male KM mice pretreated with or without PMFs were anesthetized and received abdominal γ-ray irradiation at a single dose of 12 Gy to establish a radiation-induced intestinal injury mice model (Figure 4A). The mice were kept in an experimental animal barrier environment in the following 7 days after irradiation. No mice were found dead during the observation period. Medium dose (200 mg/kg) treatment significantly reduced body weight loss after irradiation, while high dose (800 mg/kg) and low dose (50 mg/kg) treatments showed no benefits in body weight maintenance (Figure 4B). The extent of intestinal damage was preliminarily evaluated by comparing the length and the weight of the small intestine and colon (Figure 4C,D). Irradiation significantly increased the weight to length ratio both in small intestines and in colons; however, intake of PMFs caused no differences in this index. Although no significant differences in intestinal weight to length ratio were found between mice treated with different doses of PMFs, intestines from mice administrated with 50 mg/kg PMFs per day showed more severe injury, which was manifested as congestion and edema (Figure 4C).

As injury might occur in multiple organs when experimental animals are exposed to abdominal irradiation, organ indexes of the liver and spleen were also calculated to determine the protective effects of PMFs on multiple organs. Both the liver index and spleen index of mice in the VEH group significantly changed after irradiation, while intake of 200 mg/kg sour orange PMFs per day, to some extent, reversed such trends of organ index changes. Interestingly, oral administration of a high dose of PMFs showed opposite effects on liver and spleen indexes, while the low dose of PMFs intake did not show any influence on organ indexes (Figure 4D). Mice epididymal adipose was separated and weighed, which revealed changes in body composition (fat mass) as the reason for the reduction in body weight loss brought by PMF administration. Irradiation led to a significant decrease in fat index. Intake of PMFs tended to relieve fat mass loss in mice; however, only those pretreated with 200 mg/kg PMFs showed a significant increase in the fat mass index, while 800 mg/kg or 50 mg/kg pretreatment only contributed slightly to remission with no significant differences compared to VEH group (Figure 4E).

As oral administration of 200 mg/kg sour orange PMFs positively promoted irradiation-induced injury repair, different section staining methods were used for further analysis of the protective effects of PMFs from a pathological perspective. H&E staining showed that high-dose abdominal irradiation shortened intestinal villus and caused infiltration in crypts and lamina propria in the ileum. Medium-dose PMF pretreatment, to some extent, maintained relatively normal small intestinal histological structures by keeping the integrity of villus and crypts. In the region of the proximal and distal colon, administration of 200 mg/kg PMFs mitigated loss of crypts and infiltration (Figure 5A). To determine the change in the cellular composition of bowel tissues, paraffin slices were stained with Alcian Blue (AB) and positive staining cells were regarded as secretory cells, for example, goblet cells. The number of goblet cells showed no differences between groups of mice in the small intestine; however, AB positive staining cells were observed to increase after irradiation in the proximal and distal colon. The colonic goblet cells of PMF-treated groups showed a similar phenotype to mice that received no irradiation (Figure 5B). As crypt loss and aberrant crypts were observed after irradiation, Olfm 4 (a widely used marker of intestinal stem cells) and Ki-67 (a protein expressed only during the active phases of the cell cycle) were detected to determine the influence on intestinal stem cell (ISC) maintenance and cell proliferation brought by PMFs intake. The IHC staining results of Olfm 4 showed that PMF pretreatment lessens the loss of ISCs after irradiation. The cell proliferation showed no significant differences among various groups (Figure 5B).

### 2.5. In Vitro Cytotoxicity Test of PMFs

Cytotoxicity of PMFs was tested through a CCK8 method. Two cell lines, CCD 841 CoN (normal colonic epithelium cell line) and HCT-116 (colorectal cancer cell line), were used to determine the influences of PMFs on cell viability. As PMFs started to crystalize in complete medium at a concentration of 100–120 μg/mL, concentration gradient of PMFs was set as 0, 5, 10, 20, 40, 60, 80, 100 and 120 μg/mL. The IC_50_ values of PMFs in CCD 841 CoN and HCT-116 were 42.23 μg/mL and 36.00 μg/mL, respectively. The cytotoxicity of three major monomers in sour orange PMFs was also tested using cell line CCD 841 CoN. Nobiletin was shown to crystalize in complete medium at 198 μM (80 μg/mL), and the IC_50_ was calculated as 128.18 μM (51.58 μg/mL). The crystallization concentrations of tangeretin and sinensetin were 53 μM (20 μg/mL) and 214 μM (80 μg/mL). The inhibition of cell viability brought by tangeretin and sinensetin was no more than 50% under their saturated concentration (Figure 6).

## 3. Discussion

The gastrointestinal form of acute radiation sickness (ARS) or acute radiation enteritis (RE) is a common complication of intraperitoneal radiotherapy which affects the postoperative life quality of patients and sometimes causes mortality. Studies have been carried out to explore approaches to alleviate the symptoms of ARS [17,18], including nutrition intervention, pharmaceutical discovery, daily nursing care and substitutive medication [5,6,14,17]. However, no certain cure has been put forward to solve the problems brought by ARS, and thus the exploration of ARS remains incomplete. The present study extracted and detected PMF contents in 12 sour orange materials to uncover the existing law of these highly methoxylated flavones. The PMF extract was then separated and purified by macroporous resins to remove unexpected components of flavones and flavanones (or their glucoside forms). The potential of PMFs as radio-resistant agents was evaluated by a combination of network pharmacology and bioinformatics methods. To detect the protective effects on RE, the purified PMFs were provided to experimental mice with different doses (0, 50, 200, 800 mg/kg/day). In vivo experiment showed that 200 mg/kg pretreatment mitigated the irradiation-induced injury by reducing body weight loss and maintaining the structure of the gut. Finally, cytotoxicity of PMFs and major monomers was determined by the CCK8 method by which the IC_50_ concentrations of PMFs and nobiletin were calculated as 42.23 μg/mL and 51.58 μg/mL, respectively.

Sour orange (*Citrus aurantium* L.) is important germplasm often used as rootstock and crude material of herbal medicine with a long history of application. It has been proven that sour orange peel contains various bioactive compounds, such as synephrine and rutinum [11]. As a kind of secondary metabolites, PMFs are defined as flavones bearing two or more methoxy groups (-OCH_3_) which mainly exist in citrus fruit and vegetative tissues [19]. The content and types of PMFs may be regulated by both environmental factors (biotic or abiotic factors) and the genetic backgrounds of plant materials [20]. Thus, the variation of PMFs may reveal ecological adaptability on the one hand; on the other hand, the diversity of PMFs matches well with phylogenetic relationships among taxa in genus *Citrus*. Our results demonstrated that the content and component of PMFs showed significant variation in different sour orange cultivars. The differences observed may be caused by the natural selection that occurred during the process of evolution. As all sour orange materials were collected at the same time and on the same field, which limited the environmental factor changes, variation of PMF content was possibly brought by genetic factors rather than epigenetic factors. Since Scora (1975) [21] and Barrett (1976) [22] described the taxonomy of genus *Citrus* based on components of essential oil, phytochemicals have been considered as parameters for plant taxonomic classification. In other words, closely related varieties should have similar content or component of phytochemicals.

To determine whether the variation of PMFs intraspecies matched the phylogenetic relationships of *Citrus*, we used the hierarchical clustering method for further analysis of the diversity and similarity of PMFs in sour orange. Pomelo, sweet orange and mandarin were introduced as outgroups. Based on molecular biological evidence, sour orange is considered to be a hybrid origin taxon, and pomelo is proven to be one of the crossing ancestors (maternal ancestor). While pomelo and mandarin are regarded as two basic independent origin species in the genus *Citrus*, sweet orange may be the backcross progeny of (pomelo × wild orange A) × wild orange B. Our clustering results agreed with the phylogenetic relationships among four citrus types demonstrating that PMFs have high homogeneity within taxa *Citrus aurantium* L. and differ from other species in the genus *Citrus*.

Natural components play important roles in human health, but it challenges researchers that a great variety of chemicals with different structures lead to extremely complex bioactivities. With the development of computer algorithms and databases, network pharmacology has been released for target prediction and drug discovery. In the present study, the potential of PMFs as radio-resistance agents was predicted by calculating targets enriched in pathways influenced by irradiation, which was similar to a method known as CIPHER developed previously [23]. Network pharmacology has been proven to be efficient in the investigation of traditional herbal medicine, especially in critical component mining or mechanism exploration. However, network pharmacology is highly dependent on databases, so the computing method may not apply to investigating new-found chemicals or chemicals that have received little attention. Besides, in the course of data collection, errors of omission and duplication occur frequently [24]. Zhou et al. [25] used the network pharmacological method to study the effects and mechanism of xuebijing (a traditional Chinese patent medicine for sepsis). They found that targets downloaded from databases only covered 80–90% of targets reported in the literature, demonstrating the limitation of network pharmacology. Thus, wet experiments are required for further investigation of natural compounds.

The intestine, regarded as an important digestive and immune organ, has complete strategies of cell differentiation and proliferation to meet the requirement of digestion and resistance to external threats. Various intestinal cells and complex intestine microecology guarantee the homeostasis of the gut and, at the same time, provide support to intestinal barrier function [26]. Acute radiation enteritis is featured as a disease accompanied by hematochezia, mucous edema, ulcer and diarrhea (or constipation). Mechanical barrier function loss is one of the symbols of ARS. Mechanical barrier function loss is on the one hand due to increased apoptosis of epithelial cells and loss of intestinal stem cells caused by irradiation; on the other hand, it agrees with the downregulation of tight junction proteins (TJPs) [7]. Several pathways have been proven to be correlated with mechanical barrier function loss, such as the p53 pathway [27], p38/MAPK pathway [28] and Rho/ROCK pathway [29]. In the present study, PMFs mitigated the crypt loss induced by irradiation and maintained relatively normal intestinal structures compared to the VEH group. PMFs have been proven to regulate the expression of TJPs in chemically induced colitis and to monitor the process of autophagy. In vivo experiments have provided evidence that PMFs influenced apoptosis through the target of p53. Besides, MAPK and Rho were also reported as targets of PMFs. These studies provided the potential mechanisms by which PMFs alleviate mechanical barrier function loss.

Secretory cells, such as goblet cells, produce mucus containing lysozymes, mucopolysaccharides, glycolipids and digestive enzymes which make up the intestinal chemical barrier. Changes in the number and function of goblet cells may, to some extent, reveal the changes in the chemical barrier [30]. Colonic goblet cells protect crypts from infection by forming mucus plumes to cover the stem cell niche. A newly identified subpopulation of goblet cells (intercrypt goblet cells (icGCs)) was found to be essential for colonic mucus barrier function [31]. Loss of icGCs might lead to higher risks of colitis, demonstrating the importance of goblet cells in epithelial homeostasis [32]. The number of goblet cells is regulated by Sprouty 2, an intracellular signaling regulator, which is highly expressed in the colonic epithelium. Under the condition of colitis, the expression of Sprouty 2 will be downregulated leading to an increase in goblet cells through the PI3K/Akt pathway [33]. Therefore, we hypothesized that the increase in colonic goblet cells after irradiation might be a response to inflammation. As PMFs have been confirmed to block inflammation through the PI3K/Akt pathway [34], the intervention might regulate the number of colonic goblet cells through a similar pathway.

It was inferred that small intestinal goblet cells did not show differences between VEH- and medium-dose-treated groups because the fast-pass characteristic of contents in the ileum and relatively low microbial abundance limited the interaction between PMFs and intestine. Evidence is mounting that the gut microbiota is essential for PMF metabolism and PMFs can shape the microorganism characteristics. Modification of PMFs might be carried out by gut microorganisms, including demethylation, hydroxylation, glycosylation and ring opening [35]. Meanwhile, PMFs could influence metabolic pathways, for example, the amino acid metabolism pathway. Fecal microbiota transplantation (FMT) further proved that bacterial community structures were significantly influenced by PMF pretreatment and produced long-term effects on recipient mice [36]. There are massive and complex microbial communities colonizing the gastrointestinal (GI) tract, composing the intestinal microbial barrier. High-dose irradiation leads to disruption in the balance of microbial communities and increases the risk of severe infection [37,38]. In summary, the mechanisms of PMFs in relieving ARS are complex and multifaceted.

PMFs have been confirmed to have multiple bioactivities and, for a long time, have been considered as nontoxic agents [39]. Because of poor water solubility, PMFs may crystalize in the medium at certain concentrations. Our results showed that the inhibition of cell viability brought by tangeretin and sinensetin was no more than 50% under their saturated concentration. It should be pointed out that environments inside the bowel lumen differ from those in the medium and the effects of microorganisms should be taken into consideration. Recent studies on the toxicology of tangeretin and PMFs showed that the in vivo toxicity of PMFs appeared to be a U-shaped dose–response panel and had sexual selective effects [39,40]. These results together demonstrate that more detailed toxicological research should be done to determine the safety of the PMFs.

There are some limitations of this study. First, PMFs used in the experiment were purified from a single growth stage, and PMFs in other tissues were ignored. Further and processive analysis of the diversity of PMFs in different sour orange cultivars, the variation of PMFs during the growth stages of sour orange and the polymorphism of PMFs in different tissues of sour orange should be carried out to provide theoretical and practical guidance for the utilization of sour orange materials. Second, because the metabolites of PMFs in vivo remain unclear, only major monomers were adopted for network pharmacological analysis. Given the development of MS and computing technologies, the metabolic process of PMFs could be uncovered by new pharmacology methods, such as Poly PK analysis. Finally, the strains of mice and cell lines covered are relatively limited. Omics and genetically modified animals should be used to explore the mechanism of PMFs as radio-resistant agents.

## 4. Materials and Methods

### 4.1. Plant Materials

Sour orange fruits of 12 cultivars were collected from National Citrus Germplasm Repository, Citrus Research Institute, SWU/CAAS, China, in December 2020. The fruits were collected following a five-point sampling method to reduce errors brought by environmental factors. The materials used are listed in Table 1.

### 4.2. PMF Extraction and Detection

The peels of sour orange were separated from flesh and immediately frozen in liquid nitrogen. The peels were then porphyrized and sieved to obtain materials for extraction. The powder-like materials were kept at −80 °C until extraction. The powders were rewarmed to room temperature 1 h before extraction. One gram of each powder was moved into a 15 mL centrifugal tube and mixed with 8 mL methanol. The mixtures were then treated with ultrasound (300 Hz, 41 °C, 34 min). The supernatants were removed and placed into 50 mL centrifugal tubes after centrifugation (1500 rpm, 5 min). Sediments were supplemented with 8 mL methanol and repetitively extracted twice. The supernatants were combined and diluted with methanol to a final volume of 25 mL. The extracts were filtered with 0.22 μm hydrophilic polytetrafluoroethylene (PTFE) syringe filters to obtain a sample solution for ultrahigh-performance liquid chromatography (UPLC, Waters ACQUITYI-Class (Milford, MA, USA) equipped with a photodiode array detector, a quaternary solvent delivery system and a column temperature controller) detection. Chromatographic separations were performed on a 2.1 × 100 mm, 1.7 µm ACQUITY UPLC HSS C18 column (Waters, Milford, MA, USA). The mobile phase consisted of water/formic acid (99.99%: 0.01%, *v*/*v*) (A) and methanol (B) at the rate of 0.4 mL/min, and the gradient profile was as a previous study reported. All the prepared samples were tested in triplicate [41]. The standard information used in our experiment is listed in Appendix A. Quantification methods (standard curves of each monomer) are listed in Appendix A. The data were visualized by OriginPro 2022 (available from https://www.originlab.com) (accessed on 15 December 2021).

### 4.3. Purification of PMFs in Selected Sour Orange Cultivar

The methanol extract of the selected cultivar was purified by a dynamic adsorption–desorption method established previously [42]: a laboratory-scale glass column (300 mm × 22 mm i.d.) packed with 11.0 g wet HPD 300 resins (Cangzhou Bon Adsorber Technology Co., Ltd., Cangzhou, Hebei, China). The extracts were carefully loaded onto the resin column at feeding speeds of 4 BV/h, and the loading concentration of total PMFs was approximately 6 mg/mL. The resins were washed with 7% ethanol to remove water-soluble matter. Ninety percent ethanol was used for the desorption of PMFs at a speed of 4 BV/h until the content of PMFs in the eluent was less than 10% of the loading mass.

The eluent was concentrated by a rotary evaporator and then placed at 4 °C for one week to enhance the process of crystallization. The crystals of the flavonoid mixture were collected and resolved in methanol. The solution was then detected by UPLC.

### 4.4. Bioinformatics and Network Pharmacology Analysis of Mice ARS

The SRA data of dataset GSE126507 were downloaded from NCBI and then transferred into fastaq form using SRA Toolkit. The visualization of data in this part was carried out by R Version 3.6.3 (R Core Team (2020). R: A language and environment for statistical computing. R Foundation for Statistical Computing, Vienna, Austria). The differentially expressed genes were obtained by Deseq2 package. The heatmap was plotted by the ComplexHeatmap package. The enrichment of differentially expressed genes was carried out using the ggplot2 package and GOplot package.

The potential targets of PMFs were downloaded from online databases SWISS and SEA. The downloaded targets were merged and duplicates were removed, followed by GO/KEGG enrichment using ggplot2 and GOplot packages. The protein–protein interaction (PPI) network was established by STRING and modified by Cytoscape Version 3.8.2 (National Institute of General Medical Sciences, USA).

### 4.5. In Vivo Experiments

Eight-week-old KM male mice weighing 28−30 g were purchased from Hunan SJA Laboratory Animal Co., Ltd. (Changsha, China) (production certificate No. SCXK (Xiang) 2019-0004), and kept in the laboratory animal room of the Institute of Combined Injury, AMU. Mice were randomly divided into 6 groups (*n* = 5): Blank, mice received no PMFs or irradiation; VEH, mice received no PMFs but were injured by irradiation; Low, mice received 50 mg/kg PMFs per day and were injured by irradiation; Medium, mice received 200 mg/kg PMFs per day and were injured by irradiation; High, mice received 800 mg/kg PMFs per day and were injured by irradiation; Drug, mice received 200 mg/kg PMFs per day but were not injured by irradiation. After a 7-day adaption, mice were irrigated with different doses of purified PMFs (0, 50, 200, 800 mg/kg) dispersed in sterile 0.8% CMC-Na solution once per day for consecutive 14 days. On the 7th day after the first irrigation, mice were anesthetized by a combination of 0.5% Zoletil-50 (Virbac) and 0.5% xylazine hydrochloride (Dunhua Shengda Animal Medicine Co., Ltd., Dunhua, China) and received abdominal γ-ray irradiation at a total dose of 12 Gy. The mice were transported to the laboratory animal room and fed normally. Mice were sacrificed on the 14th day after the first irrigation. The small intestines were divided into 5 segments numbered 1 to 5 and colons were also divided into 5 segments numbered 6 to 10. Segments 1, 3, 5, 7 and 9 were fixed with 10% formaldehyde–PBS solution while segments 2, 4, 6, 8 and 10 were treated with RNA later (R0118–100 mL, Beyotime) and kept at −80 °C. Livers and spleens were separated and weighed for the calculation of organ indexes. The formula of organ index calculation was as follows:(1)$$\mathrm{Organ}\;\mathrm{index}=\frac{\mathrm{Organ}\;\mathrm{weight}\;\left(g\right)}{\mathrm{Body}\;\mathrm{weight}\;\left(g\right)}$$

Pathological section staining followed the recommended protocols of corresponding regents, including H&E staining kit (C0105S, Beyotime), AB-PAS staining kit (G1285, Solarbio), OLFM4 (D1E4M) XP Rabbit mAb (14369, CST) and Ki-67 (D3B5) Rabbit mAb (Mouse Preferred; IHC Formulated) (12202, CST).

### 4.6. In Vitro Experiments

Cell lines of CCD 841 CoN and HCT-116 were provided by State Key Laboratory of Trauma, Burns and Combined Injury (Army Medical University, Chongqing, China). Cells were resuscitated and cultured in DEME (11965092, Gibco) containing 10% FBS (04-121-1A, BI) under the conditions of 37 °C and 5% CO_2_ for one generation and then seeded into 96-well plates (1 × 10^5^ cells per well). PMFs and monomers were dissolved in DMSO and then diluted with DMSO (472301, Sigma, Burlington, MA, USA) to obtain secondary mother liquor of different concentrations (0, 0.5, 1, 2, 4, 6, 8, 10 and 12 mg/mL). The liquor was added into the medium (1 μL per well) and incubated with cells overnight. Cell viability was detected by CCK8 kit (CCK801, Multi sciences) following the recommended protocols.

### 4.7. Statistical Analysis

Data were expressed as the means ± standard deviation of replicates. Significant differences among groups were analyzed by *t*-test (*p* < 0.05) and visualized by GraphPad Prism Version 9.0.0 for Windows (GraphPad Software, San Diego, CA, USA, http://www.graphpad.com (accessed on 20 November 2020)) and OriginPro 2022.

## 5. Conclusions

In summary, the present study analyzed PMF contents in 12 sour orange materials and determined the protective effects on ARS. The in vivo and in vitro toxicities of PMFs were analyzed to provide information for PMF development. This study provides theoretical and practical guidance for the utilization of sour orange resources.

## Figures and Tables

**Figure 1 molecules-27-01934-f001:**
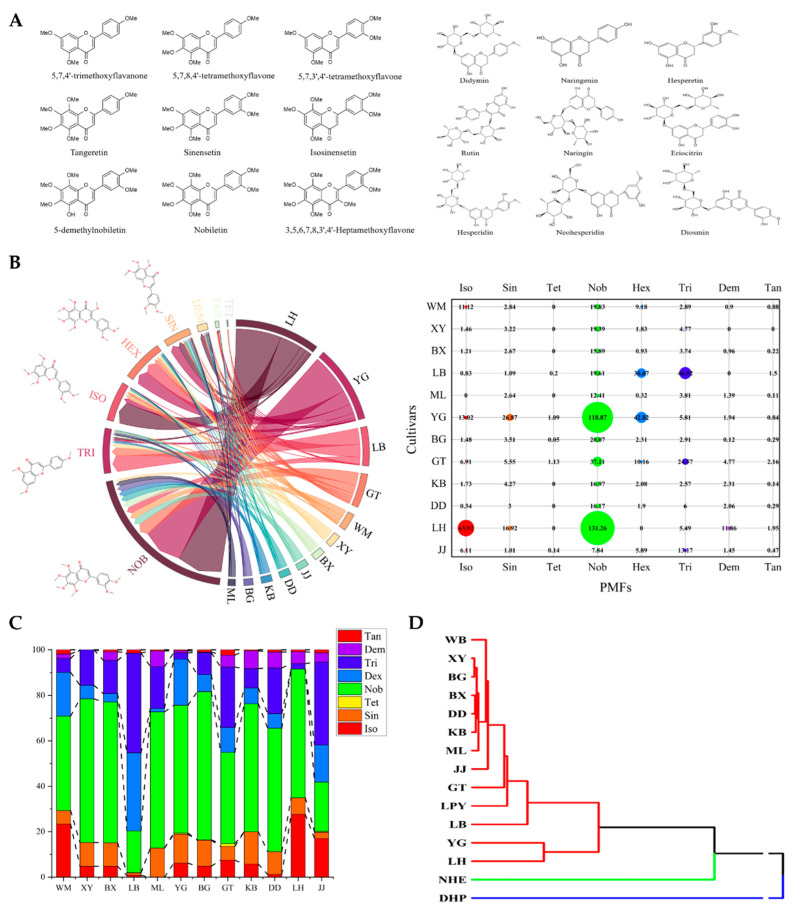
The PMF content in sour orange. (**A**) The structures and the names of flavonoid standards used in this study. (**B**,**C**) The absolute and relative content of PMFs in sour orange. (**D**) Clustering analysis of *Citrus* taxa based on PMF content.

**Figure 2 molecules-27-01934-f002:**
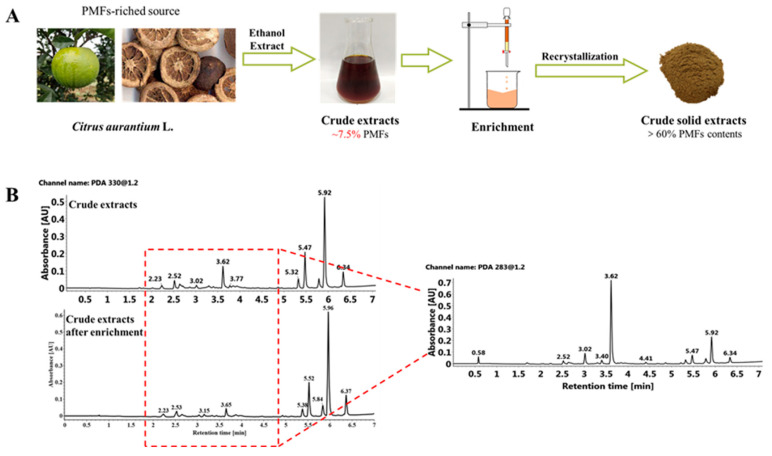
Purification of PMFs in sour orange. (**A**) The experimental design of the purification of PMFs in sour oranges. (**B**) UPLC chromatograms of PMFs before and after purification.

**Figure 3 molecules-27-01934-f003:**
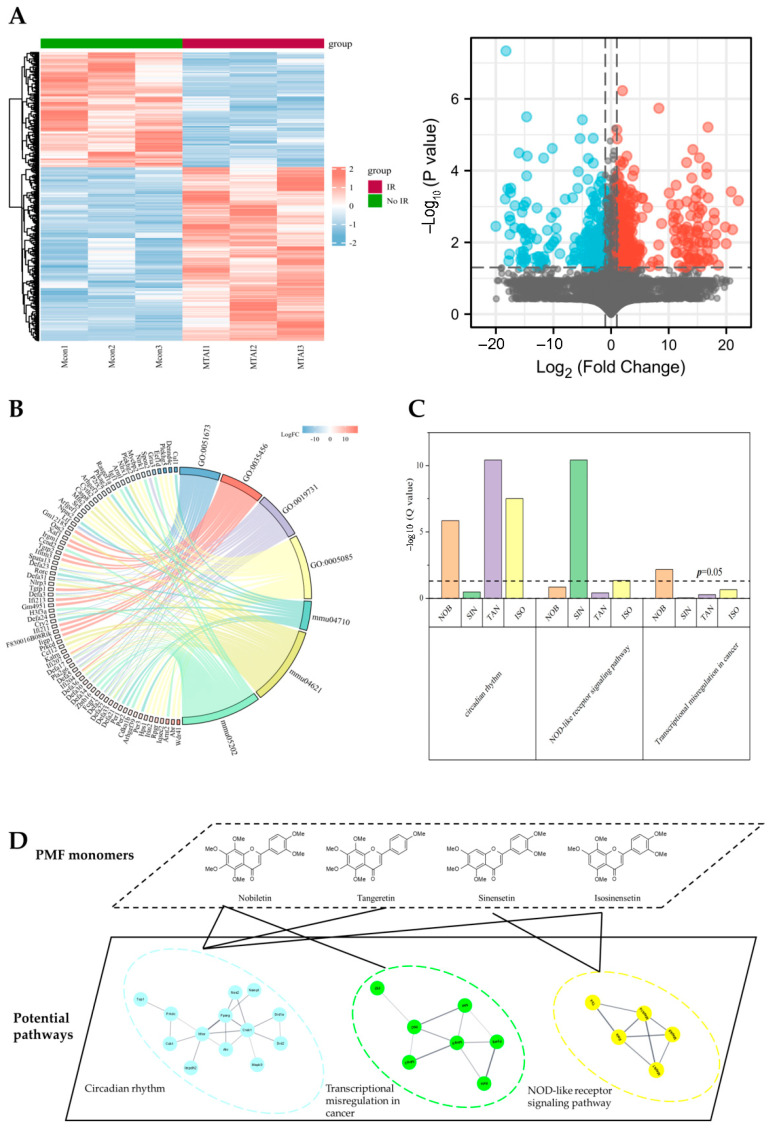
Bioinformatics and network pharmacology analysis of ARS and PMF targets. (**A**) Heatmap and volcano plot of differential expression genes after irradiation. (**B**) Chord diagram of GO/KEGG enrichment results and related genes. (**C**) Enrichment analysis of each monomer’s targets. (**D**) The compound–biological functional module–molecule network of PMFs.

**Figure 4 molecules-27-01934-f004:**
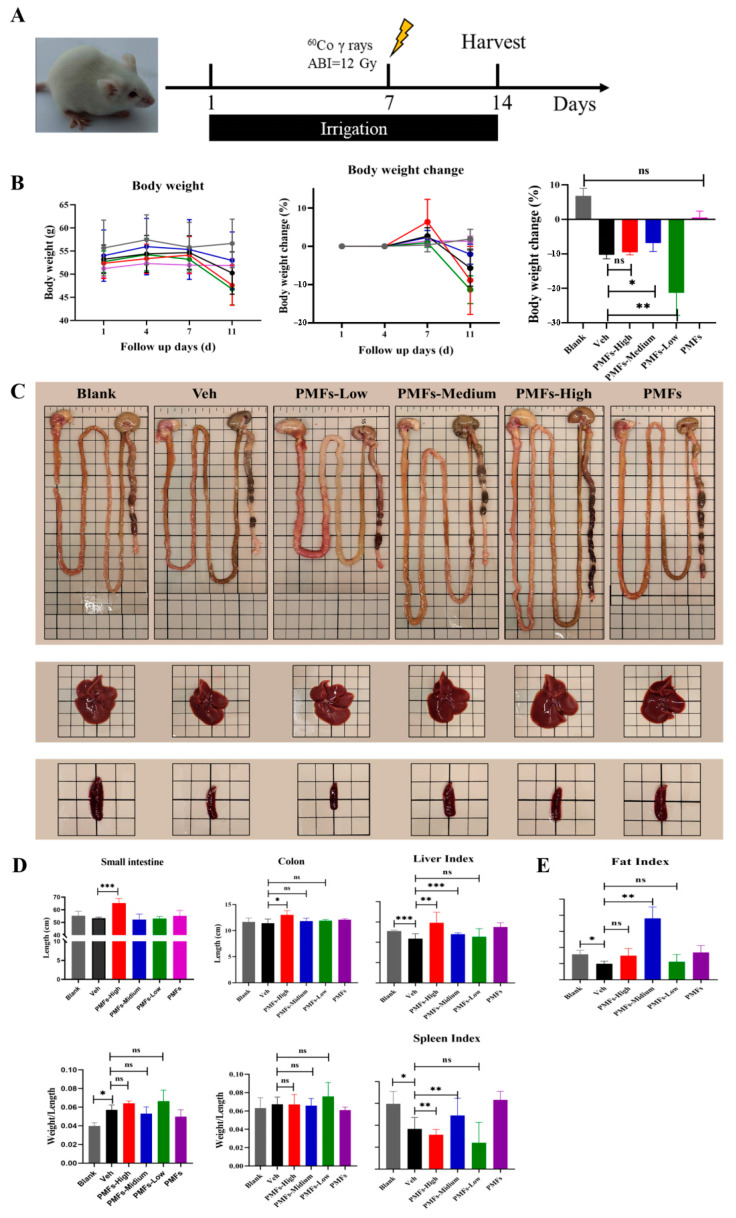
The protective effects of PMFs on irradiation-induced injury. (**A**) Experimental design. (**B**) Body weight change of mice during the experiment. (**C**) Major intraperitoneal organs of differently treated mice. (**D**) Organ indexes of mice. (**E**) Fat mass index of mice. * *p* < 0.05; ** *p* < 0.01; *** *p* < 0.001. ns, not significant.

**Figure 5 molecules-27-01934-f005:**
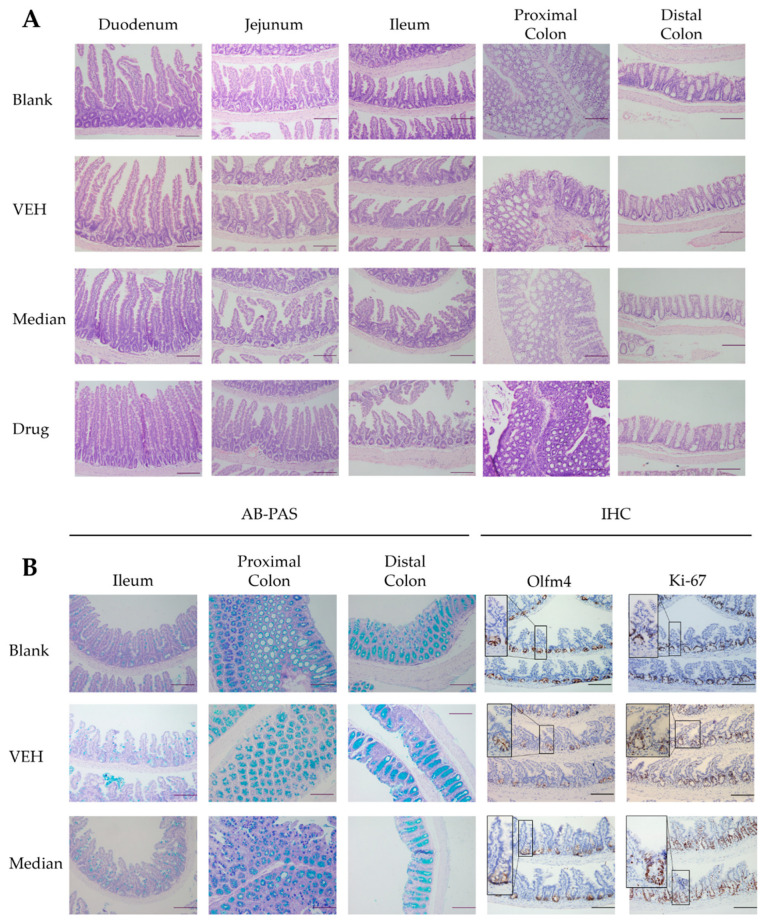
Staining of pathological sections. (**A**) H&E staining. (**B**). AB-PAS staining and IHC staining of Olfm 4 and Ki-67.

**Figure 6 molecules-27-01934-f006:**
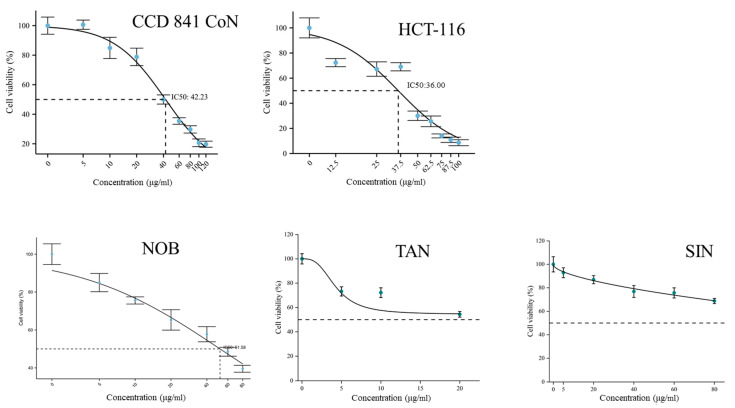
Influences on cell viability caused by PMF treatment.

**Table 1 molecules-27-01934-t001:** List of sour orange resources collected in Citrus Research Institute, SWU/CAAS.

Germplasm Code ^1^	Germplasm Name ^2^	Place of Origin	Abbreviation
LA0002	Jiangjin suancheng	China	JJ
LA0003	Daidai	China	DD
LA0004	Hangbulaite suancheng	U.S.A.	KB
LA0008	Goutoucheng	China	GT
LA0012	Banggan	China	BG
LA0015	Yaogan-1	China	YG
LA0018	Moluogesuancheng	Morocco	ML
LA0022	Lubidukesi suancheng	Mexico	LB
LA0038	Baxi suancheng	Brazil	BX
LA0048	Xiaoye suancheng	U.S.A.	XY
LA0055	Lianhe suancheng	China	LH
LA0057	Wanmucheng	Japan	WM

^1^ Germplasm code is a unique number representing a certain germplasm in National Citrus Germplasm Repository, Citrus Research Institute, SWU/CAAS, China. ^2^ Germplasm name refers to the registration name which can be retrieved from the National Citrus Germplasm Repository website of Citrus Research Institute, SWU/CAAS, China.

## Data Availability

A publicly available dataset (GSE126507) was analyzed in this study. This data can be found here: https://www.ncbi.nlm.nih.gov/geo/query/acc.cgi?acc=GSE126507 (accessed on 5 January 2022).

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
