# Peer review of "The Protective Effects of Sour Orange (Citrus aurantium L.) Polymethoxyflavones on Mice Irradiation-Induced Intestinal Injury"

_molecules, 2022, doi:10.3390/molecules27061934_

Round 1

Reviewer 1 Report

Generally, this is a clearly written manuscript. However, there are some questions need to clarify.

  1. Line 96, check the data (varied from 28.87 to 1.07 μg/g●FW?).
  2. Check Fig. 2B. Why is the pattern of UPLC chromatograms not similar between the right and left figures? Why does the peak height of absorbance at 5.92 min retention time become much lower than that at 3.62 min retention time?
  3. Line 174, please indicate how to obtain liver and spleen index.
  4. Figure 4B, check the data about body weight and body weight change. The body weight and body weight change (%) between two figures can’t match well.
  5. Line 430, check 14 day or 7 day after the first irradiation?
  6. Please indicate how the statistical analysis was performed.
  7. The reference cited should follow the official requirement.
  8. Tiny figures make it difficult to read.

Author Response

Reviewer 1

Dear reviewer,

Thank you for your questions and suggestions on this manuscript. The mistakes were corrected in our manuscript (by revision mode) and replies to the questions and suggestions mentioned were listed as follows:

  1. Line 96, check the data (varied from 28.87 to 1.07 μg/g●FW?).

Thank you for pointing out the mistake. The sentence was rewritten as “sinensetin (varied from 28.87 μg/g•FW to 1.01 μg/g•FW among cultivars)” (Line 101). We have checked the data carefully to ensure the accuracy of our manuscript.

  1. Check Fig. 2B. Why is the pattern of UPLC chromatograms not similar between the right and left figures? Why does the peak height of absorbance at 5.92 min retention time become much lower than that at 3.62 min retention time?

Thank you for your questions. In Figure 2B, the upper and lower figures on the left were the chromatograms of sour orange PMFs ethanol extracts before and after purification, respectively. The absorbance peaks were detected at a wavelength of 330 nm where PMFs showed higher peaks. The figure on the right was the absorbance peaks of components in the wash (7% ethanol-water solution) detected at a wavelength of 283 nm where flavanones showed higher peaks. The whole figure 2B showed the efficiency of our purification strategy by detecting both the eluent and the wash by UPLC. The reasons for the differences showed among figures were listed as follows:

  • We used chromatograms detected at different wavelengths to show the differences between the eluent (PMFs-enriched extracts) and the wash (consisted largely of other flavonoids), which led to the dissimilarity between the right and left figures.
  • It has been proven in our previous study that the optimum absorbance wavelengths of PMFs (330 nm) and flavanones (283 nm) differ (Li et al. Simultaneous Separation and Purification of Five Polymethoxylated Flavones from “Dahongpao” Tangerine (Citrus tangerina Tanaka) Using Macroporous Adsorptive Resins Combined with Prep-HPLC, Molecules, 2018, 23, 2660; doi:10.3390/molecules23102660). We provided the absorbance peaks of flavanones in figure 2B at 330 nm and 283 nm to better display the differences between the components in the eluent and the wash which led to dissimilarity between figures on the right and on the left.
  1. Line 174, please indicate how to obtain liver and spleen index.

Thank you for your suggestion. The formula on how to calculate the liver and spleen index was now added in Line 445-448 “Livers and spleens were weighed for the calculation of organ indexes. The formula of organ indexes calculation was listed as following: Organ index= Organ weight (g) /Body weight (g)”.

  1. Figure 4B, check the data about body weight and body weight change. The body weight and body weight change (%) between two figures can’t match well.

Thank you for your reminder. We have carefully checked the data we used in figure 4B. In the curves of body weight (the figure on the left), we traced the exact body weight (g) on different days. In the curves of body weight change (%), the data we used were obtained from the following formula:

Thus, the 2 curves might look different but they matched with each other.

  1. Line 430, check 14 day or 7 day after the first irradiation?

Thank you for pointing out our mistake. We apologize for the wrong information provided in our manuscript. We corrected the sentence as “Mice were sacrificed on the 14th day after the first irrigation”.

  1. Please indicate how the statistical analysis was performed.

Thank you for your suggestion. We added statistical analysis method in the part of “Materials and methods”:

“4.7 Statistical Analysis

Data were expressed as the means ± standard deviation of replicates. Significant differences among groups were analyzed by t-Test (p < 0.05) and visualized by GraphPad Prism (Ver 9.0.0).” (Line 463-466)

  1. The reference cited should follow the official requirement.

Thank you for your suggestion. We checked the requirements on reference in the official templet and our manuscript. We found that references cited in our manuscripts followed the format of “Author 1, A.B.; Author 2, C.D. Title of the article. Abbreviated Journal Name Year, Volume, page range” or “In the text, reference numbers should be placed in square brackets [ ], and placed before the punctuation; for example [1], [1–3] or [1,3]”. The references were inserted by “molecules” style in Mendeley (Ver 2.66.0).

  1. Tiny figures make it difficult to read

Thank you for your suggestion. We have improved the resolution of the figures in the manuscript and slightly adjusted the sizes of the figures. Moreover, we uploaded high-resolution, original figures with our submission.

Reviewer 2 Report

ABSTRACT

Gy:  what is the meaning?

Lines 59-61: A variety of bioactive compounds were detected from sour orange, such as flavonoids, volatile oil, limonoids and alkaloids.

Rewrite that sentence. Oils are not made up of a single class of natural products, as is the case with the other classes mentioned above.

RESULTS

ITEM 2.1.

  • Make it clear! Which method did you employed for PMFs detection? UPLC with standards? If so, I believe it is very important to detail the acquisition of samples, the solutions preparation and their quantification methods;
  • What kind of analysis was done to generate figure 1B? Be clearer! I would like to interpret the figure on my own. I make the same observation for 1C;
  • I believe that the qualitative and quantitative differentiation of PMFs in the different cultivars would be more appropriate if it had been performed by chemometric approaches (PCA, among others).

Figures 1, 3 and 4: the figures are too small. This makes it difficult to visually perceive the results.

4.1 Plant materials (Table 1), column (place of origin)

- What you mean when you say AMERICA? be more specific. America has many countries.

Author Response

Dear reviewer,

We truly appreciate all your advice and suggestions on our manuscript. The mistakes and inappropriate presentations were corrected in our revised version. Detailed changes are as follows.

ABSTRACT

  1. Gy: what is the meaning?

Thank you for your question. “Gy” is the abbreviation for “Gray”, a SI* derived unit of radiation dose, which is often used to quantify the absorbed doses of items. According to the conception, “1 Gy” means that 1 joule of radiant energy is absorbed by 1 kilogram of material.

Gray’s definition is

*SI is the abbreviation for international system of units.

  1. 2. Lines 59-61: A variety of bioactive compounds were detected from sour orange, such as flavonoids, volatile oil, limonoids and alkaloids.

Rewrite that sentence. Oils are not made up of a single class of natural products, as is the case with the other classes mentioned above.

Thank you for your advice. The sentence was rewritten as “A variety of bioactive compounds were detected from sour oranges, such as flavonoids, volatile components (i.e. limonene, pinene, Linalool, etc.), and alkaloids”. (Line 59-61)

RESULTS

  1. ITEM 2.1.
  • Make it clear! Which method did you employed for PMFs detection? UPLC with standards? If so, I believe it is very important to detail the acquisition of samples, the solutions preparation and their quantification methods;

Thank you for your suggestion.

  • We added the methods of PMFs detection in the part “2 PMFs extraction and detection”. Briefly, “Chromatographic separations were performed on a 2.1 × 100 mm, 1.7 µm ACQUITY UPLC HSS C18 column (Waters, MA, USA). The mobile phase consisted of water/formic acid (99.99%: 0.01%, v/v) (A) and methanol (B) at the rate of 0.4 mL/min, and the gradient profile was as a previous study reported. All the prepared samples were tested in triplicate”. See line 393-398.
  • Yes, we used 18 flavonoids standards for PMFs detection. The acquisition of samples and the solutions preparation process was described in line 380-389. The standards information used in our experiment was listed in Table S2. Quantification methods (standard curves of each monomer) were listed in Table S3. (Line 398-400)
  • What kind of analysis was done to generate figure 1B? Be clearer! I would like to interpret the figure on my own. I make the same observation for 1C;

Thank you for your advice and we apologize for our unclear description. We added the analysis method that we used to generate figure 1B. The sentences consist of lines 89-91, as “The exact contents of monomers in each cultivar were calculated and summated to analyze the PMFs distribution patterns in various sour orange cultivars from the component level”. To distinguish figure 1B and figure 1C, we also added “To determine the ratio of PMF monomers in sour orange cultivars, a 100% stacked bar chart was generated to show the PMFs distribution patterns from the cultivar level” in line 96-98.

  • I believe that the qualitative and quantitative differentiation of PMFs in the different cultivars would be more appropriate if it had been performed by chemometric approaches (PCA, among others).

Thank you for your suggestion. An PCA was carried out and the results of PCA was affiliated to our manuscript as Figure S1.

  1. Figures 1, 3 and 4: the figures are too small. This makes it difficult to visually perceive the results.

Thank you for your suggestion. We have adjusted the sizes of figures in our revised manuscript.

4.1 Plant materials (Table 1), column (place of origin)

- What you mean when you say AMERICA? be more specific. America has many countries.

Thank you for pointing out our mistakes. We have changed the word “America” to “U.S.A.” for the accuracy of our materials.
